



# A novel rotor blade fatigue test setup with elliptical biaxial resonant excitation

David Melcher[1], Moritz Bätge[1], Sebastian Neßlinger[2]

[1]Divison Structural Components, Fraunhofer Institute for Wind Energy Systems, Am Seedeich 45, 27572 Bremerhaven, Germany
[2]Nordex Energy GmbH, Langenhorner Chaussee 600, 22419 Hamburg, Germany

*Correspondence to*: David Melcher (david.melcher@iwes.fraunhofer.de)

**Abstract.** The full-scale fatigue test of rotor blades is an important and complex part of the development of new wind turbines. It is often done for certification according to the current IEC (2014) and DNV GL AS (2015) standards. Typically, a new blade design is tested by separate uniaxial fatigue tests in both main directions of the blade, i.e. flapwise and lead-lag. These tests are time consuming and rather expensive due to a high number of required load cycles, up to 5 million. Therefore, it is important to run the test as efficiently as possible. During fatigue testing, the rotor blade is excited at or near its resonant frequency. The trend for new rotor blade designs is toward longer blades, leading to a significant drop in their natural frequencies, and a corresponding increase in test time. In order to reduce the total test time, a novel test method aims to combine the two consecutive uniaxial fatigue tests into one biaxial test. The biaxial test excites the blade in both directions at the same time and at the same frequency, resulting in an elliptical deflection path of the blade axis. Using elliptical loading, counting of damage equivalent load cycles is simplified in comparison to biaxial tests with multiple frequencies. In addition, the maximum loads in both main directions remain separated, while off axis loading is introduced. To achieve such a test, specific load elements need to be arranged to equalize the natural frequencies of the test setup for both test directions. This is accomplished by adding stiffness or inertial effects in a specific direction.

This work describes a new method to design suitable test setups. A parameterized finite element (FE) model of the test with beam elements for the blade represents the test setup. A harmonic analysis on the FE-model can identify the load distribution and the test conditions of a specific test setup within seconds. An optimization algorithm that varies parameters of the model and searches for the optimal setup is then applied to the analysis. This approach enables the efficient determination of a test setup, suited to the predefined requirements. The method is validated by application on three different test scenarios for a modern rotor blade: a) state-of-the-art uniaxial setups, b) uniaxial setups including springs and c) a biaxial setup. In conclusion the resulting setups are evaluated in terms of test quality and efficiency.



# 1 Introduction

Rotor blades of wind turbines are exposed to very high fatigue loading over their common lifetime of 20 years in the field. Hence, their reliability and structural integrity are very important, from both an economic and a safety perspective. When a new rotor blade design is developed, it needs to be certified before it can go into operation. The design and certification of

wind turbine rotor blades is done according to the current IEC (2014) and DNV GL AS (2015) standards. One essential part of the certification process is full-scale fatigue testing. These tests are performed to validate the calculations and assumptions made in the design models, by applying damage equivalent loads to the blade. Typically, two consecutive tests are done, in which the blade is separately loaded in the two main directions, i.e. flapwise and lead-lag. A drawback of these unidirectional tests is that the loads introduced into the blade do not necessarily represent the loading the blade will

experience under field conditions as Rosemeier et al. (2018) have shown.

The following requirements are considered in fatigue testing: A target bending moment distribution, derived from the design requirements, needs to be matched or exceeded everywhere within the area of interest, while keeping the exceedance as small as possible in order to avoid unrealistic failures. If a section of the blade is overloaded it can be damaged before the rest of the blade has been fully tested. In order to safely proceed testing, these damages need to be repaired, which can lead

to prolonged testing times. Besides the required load distribution, a high test-frequency is desired to reduce the total testing time. The energy consumption of the test is another important boundary condition. Moreover, the introduction of high local shear forces to the blade should be avoided.

A state-of-the-art fatigue test setup involves the application of different load elements, i.e. actuators and masses, along the span of the blade, while the root of the rotor blade is attached to a test block. The actuators excite the blade at or near the

corresponding system resonant frequency, while the masses are attached to the blade to tune the bending moment distribution. Forcing the blade to oscillate at a frequency significantly outside the resonant frequency leads to high shear forces at the load introduction and high energy-consumption of the test.

Attempts have been made to improve these uniaxial full-scale fatigue tests. Lee and Park (2016) for example used an algorithm to optimize the overloading by determining the optimal mass distribution, actuator position and excitation

frequencies. Similarly, Zhang et al. (2018) evaluated a different optimization algorithm and included the position of blade tip cut off as a design variable. Eder et al. (2017) proposed a uniaxial multi-frequency approach to replicate the actual spatial damage distribution of the blade more realistically.

Combining the two uniaxial tests into one biaxial test, which implies exciting the blade in both directions at the same time, is an approach to save testing time. It also has the potential to emulate the comprehensive damage along the blade's

circumference (Heijdra et al., 2013). Exciting the blade at two different resonant frequencies, resulting in a random phase biaxial fatigue test, has already been researched for a number of years (White, 2004; White et al., 2005; Greaves et al., 2012; Snowberg et al., 2014). Another approach for biaxial testing is the phase locked excitation where the excitation frequencies in flapwise and lead-lag direction have a specific ratio, e.g. 1:1 or 1:2 (White et al., 2011). Heijdra et al. (2013) proposed to





excite the flapwise mode in its natural frequency while forcing the lead-lag motion in the same frequency to get a 1:1 ratio. A method to achieve the desired frequency ratio while exciting both directions in resonance is to tune both natural frequencies independently with the concept of virtual masses and spring elements (Post, 2016). These elements are connected to the blade in such a manner, that they only add inertia or stiffness in one direction.

This work focuses on the phase locked resonant biaxial excitation with a frequency ratio of 1:1. This way each cross section along the blade axis follows an ellipse as depicted in Fig. 1(b). This simplifies counting of damage equivalent load cycles when compared to biaxial tests with multiple different frequencies. To tune the natural frequencies to be equal or very close to each other, i.e. raising the flapwise and/or lowering the lead-lag natural frequency, virtual masses and spring elements are used as Post (2016) suggested. This results in a test setup as indicated in Fig. 1(a). Unlike Post (2016), who simulated the

two test directions independently, this work aims to consider coupling of the two directions. Instead of two separate two-dimensional simulations one three-dimensional simulation is performed. Here, the virtual masses and spring elements are not just applied in the global flapwise or lead-lag direction, but tilted around the blade axis as described below. The method elaborated in the following was developed to optimize these biaxial fatigue test setups. Thus, it is used to identify which load element needs to be placed where along the blade, in order to generate a fatigue test that satisfies all test requirements as

efficiently as possible.

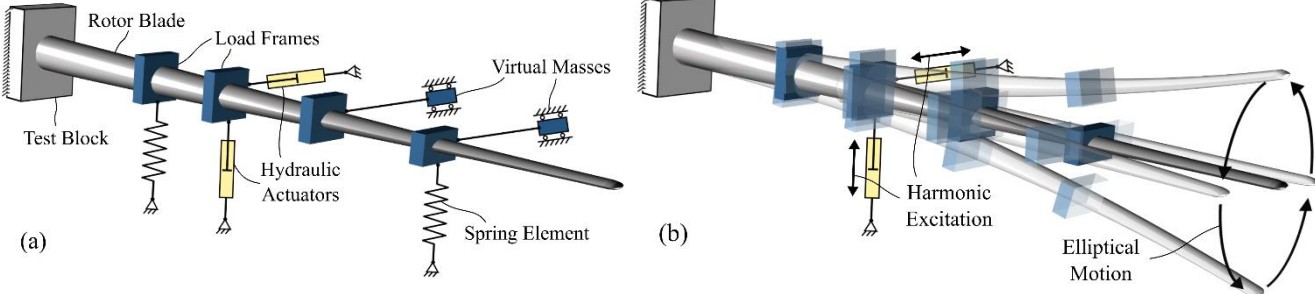

**Figure 1: Biaxial fatigue test: (a) schematic of test setup, and (b) elliptical resonant excitation and resulting displacement.**



## 2 Fatigue test simulation

To find a suitable test setup for either biaxial or uniaxial fatigue tests it is necessary to evaluate any given setup. A parameterized FE simulation tool was developed in ANSYS APDL to do so. Using specific design parameters, any desired test setup can be modelled with FE-elements. Subsequently the developed tool runs various analyses (modal and harmonic)
on the model to evaluate the properties of the given setup. This process is described in the following sub-sections.

### 2.1 Test setup modelling

The rotor blade is modelled using beam elements that are based on the Timoshenko beam theory. The properties of those beam elements consider coupling terms between different generalized strains and loads as they occur in composite structures. These properties are derived from preceding analyses of multiple different cross sections along the blade.
The model of the blade is constrained at the root in all degrees of freedom to generate a cantilever beam setup. The loading elements, which are controlled by various parameters, are then attached to the beam elements as described below. Damper elements are also attached in the model representing the aerodynamic drag that occurs during testing.

#### 2.1.1 Loading elements

Masses attached to the blade using load frames are applied directly to the beam model using point mass elements.
The load elements, which are designed to affect either the first flapwise or the first lead-lag frequency, can be virtual masses or spring elements. The virtual masses are modelled as mass elements that are connected to the beam elements using rigid body elements. They are constrained as shown in Fig. 2(a). The springs are modelled using linear spring elements.
These load elements must be attached in a specific direction; as a rotor blade oscillates in one of its first mode shapes, the direction of displacement is not necessarily aligned with the main directions of the blade. Additionally, the mode shape
direction changes along the length of the blade, meaning every cross section may oscillate in a different direction. The effect of an element on the eigenfrequency, which is not to be affected, shall be minimized. Hence, it was found that the elements need to be attached perpendicular to the mode shape of the rotor blades, which is not to be affected. This is shown in Fig. 2(a) for one representative cross section, where a spring is applied in the flapwise direction and a virtual mass in the lead-lag direction.
The mode shapes of the test setup including springs and masses differs from those of the bare blade. As the hydraulic actuators are supposed to excite the test setup with the lowest possible energy consumption, they need to be attached in line with the direction of these new mode shapes. This is visualized for a biaxial test in Fig. 2(b). For a uniaxial test, only one cylinder is attached in the respective direction.



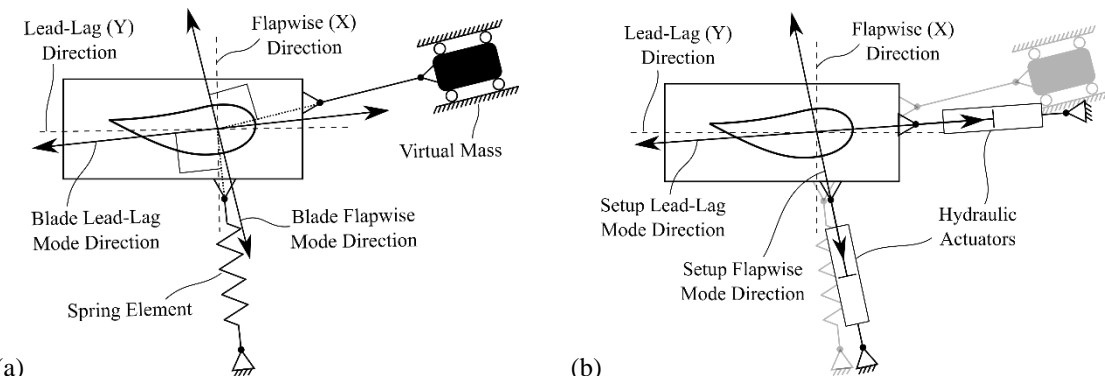

**Figure 2: Angle for application of load elements: (a) passive elements according to blade mode shape, and (b) actuators according to setup mode shape.**

For the biaxial test, the elliptical motion of the blade consists of two harmonic oscillations; one in the flapwise direction and one in the lead-lag direction, with a phase shift. This phase shift defines the tilt of ellipse around the pitch axis and the width of the ellipse. With a phase shift of 90° the ellipse is not tilted, meaning the extreme loads in one main direction do not overlap with the extreme loads of the other direction, but rather with its mean value. The phase angle needs to be controlled during testing therefore, the hydraulic actuators need to be attached at the same position along the blade length. As the hydraulic actuators may not be in line with the main directions of the blade, a skew coordinate system can be derived from their orientation as shown in Fig. 3.

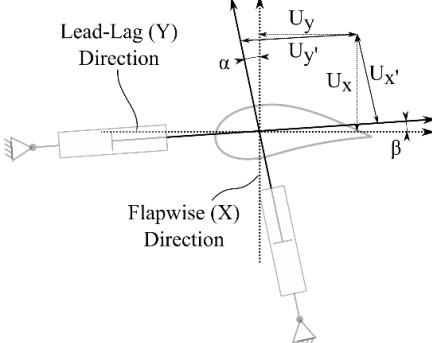

**Figure 3: Blade coordinate system and skew coordinate system of actuators.**

In order to find the correct excitation for the actuators, the desired blade motion, including the phase difference, must be converted to the skew coordinate system using Eq. (1) and Eq. (2)

$$U_{x\prime} = U_x \frac{\cos \beta}{\cos(\alpha - \beta)} - U_y \frac{\sin \beta}{\cos(\alpha - \beta)} \qquad (1)$$

$$U_{y\prime} = U_x \frac{\sin \alpha}{\cos(\alpha - \beta)} + U_y \frac{\cos \alpha}{\cos(\alpha - \beta)} \qquad (2)$$

where $U_x$ and $U_y$ are the deflection values in the blade coordinate system and $U_{x\prime}$ and $U_{y\prime}$ in the actuator coordinate system. In the real test, the angle of attack of the actuators will change constantly as the blade follows the elliptical motion. This



would result in a constantly changing skew coordinate system with moving origin. The longer the actuators are, the smaller the angle change becomes. As the harmonic simulation of the test does not consider nonlinear displacement this phenomenon has been omitted from the simulation.

### 2.1.2 Aerodynamic damping

The aerodynamic damping is modelled using linear damping elements. At each beam element of the blade, two dampers are applied; one in the flapwise direction and one in the lead-lag direction. In the simulation, each damper applies a force to the blade, which is proportional to the velocity in the corresponding direction.

Aerodynamic drag force, which corresponds to the damping force, is nonlinear and can be computed using Eq. (3).

$$F_D = -\frac{1}{2}\rho A C_D |v| v \tag{3}$$

Where $F_D$ is the drag force, $\rho$ is the density of the air, $A$ is the area perpendicular to the motion, $C_D$ is the drag coefficient and $v$ is the velocity.

To achieve the same energy dissipation with the linear damping elements as the aerodynamic drag would induce, the damping constant $C_{d,lin}$ for each element is adjusted accordingly. The theoretical formula for energy due to aerodynamic drag within one cycle of harmonic oscillation, $E_{drag}$, is shown in Eq. (4). The formula for the energy of a linear damper,

$E_{damp}$ is given in Eq. (5). When equating $E_{drag}$ and $E_{damp}$, the required damping constant can be derived as shown in Eq. (6).

$$E_{drag} = \int_0^{1/f} F_D(t)v(t)dt = \frac{16}{3}\pi^2 f^2 \hat{u}^3 C_D \rho A \tag{4}$$

$$E_{damp} = 2\pi^2 f \hat{u}^2 C_{d,lin} \tag{5}$$

$$C_{d,lin} = \frac{8}{3} f \hat{u} C_D \rho A \tag{6}$$

Where $f$ is the oscillation frequency and $\hat{u}$ is the displacement amplitude.

The drag coefficient $C_D$ used in the simulation was taken from the investigations by Greaves (2018). They derived a dependency for the drag coefficient of the displacement amplitude $\hat{u}$ and the aerofoil shape.

### 2.2 Simulation sequence

Once the test setup is modelled, including load elements and dampers, the simulation tool runs several consecutive analyses

to evaluate the given setup. Since the damping constants depend on the test frequency and the displacement amplitude, the damping constants are initially set to zero, as those variables are unknown at the beginning of the analysis. The first analysis is a modal analysis to find the natural frequencies of the test setup. The outputs are then used to find the excitation frequency for the test. For a uniaxial test, the natural frequency of the first mode in the corresponding direction is directly used as the test frequency. For a biaxial test, the mean value between first flapwise and first lead-lag frequency is used as the excitation

frequency.



The test is then simulated using a harmonic simulation with an initial actuator stroke as excitation. The harmonic simulation is then evaluated and scaled by a specific factor. This scaling factor is calculated to ensure that the test bending moment within the entire area of interest matches, or is higher than, the target bending moment to meet test requirements.

At this stage of the simulation the test frequency and a preliminary displacement amplitude is known. These are then used to update the damping constants of the damper elements; the harmonic analysis is then repeated. This iterative process of updating the damping and the harmonic analysis is repeated until convergence is achieved. This concludes the simulation and the results of the last harmonic analysis are used to evaluate the given setup.

The major advantage of using harmonic analyses in the simulation process is the short processing time. It takes only a few seconds, enabling the computation of a multitude of different test setups. A transient simulation of the test, which also considers nonlinear geometric and damping effects requires roughly 100 times more processing time. A comparison of transient and harmonic simulations yielded less than 3% difference in loading amplitudes within the area of interest.




## 3 Optimization

The process described in the chapter above can be used to evaluate any given test setup. In order to find a suitable setup, different options need to be evaluated and compared. By varying the design parameters and evaluating the influence of these changes, an optimal test setup can be found using an optimization-algorithm, as described below.

### 3.1 Problem formulation

The setups in this work are optimized to maximize the test frequency, i.e. shortest test time. This is done while keeping the ratio between test and target bending moment distribution within an allowable range. Within the area of interest the ratio must be above one to achieve the required load. To avoid overloading, an upper limit of 5% for flapwise and 10% for lead-lag loading was used. High local shear force introduction by the load elements needs to be avoided and the stroke amplitude

of the actuator needs to be kept reasonable. The design parameters that can be changed by the optimization are the mass or stiffness values of the load elements at specific defined positions along the blade. These can be attached in the flapwise or the lead-lag direction for the respective uniaxial test and in both directions for biaxial tests. Furthermore, the position of the actuators can be changed between the defined positions.

That leads to the problem formulation in Eq. (7), on which all the optimizations are based.

$$
\begin{aligned}
\text{Maximize:} \quad & \text{excitation frequency } f_{exc} \\
\text{With respect to:} \quad & \text{flapwise and/or lead lag mass or stiffness at each position} \\
& \text{actuator position} \\
\text{Subject to:} \quad & 1.00 \leq \frac{M_{x,Test}}{M_{x,Target}} \leq 1.05 \text{ within area of interest (biaxial and uniaxial flapwise)} \\
& 1.00 \leq \frac{M_{y,Test}}{M_{y,Target}} \leq 1.10 \text{ within area of interest (biaxial and uniaxial lead-lag)} \\
& 0.95 \leq \frac{f_{eig,flap}}{f_{eig,lead-lag}} \leq 1.05 \text{ (only for biaxial test)} \\
& F_A \leq 50 \text{ kN (for all load element forces)} \\
& u_A \leq 1 \text{ m}
\end{aligned}
\tag{7}
$$

### 3.2 Boundary conditions

Further assumptions and boundary conditions are applied for the simulations and the optimizations; the load frames that are needed to connect the load elements to the blade, as well as the spring elements, are assumed to be without mass.

At each position, for each main direction, the optimization can use either a mass or a spring element. It is not possible to use both at the same position for the same direction, as they would simply cancel each other out.

If a parameter set defines virtual masses in both the flapwise and lead-lag directions at the same position, the smaller mass is attached directly to the blade as load frame mass, which acts in all directions. The larger mass is reduced to the difference between the previously defined masses, as this value still needs to act in one direction.





For the uniaxial setups, no virtual masses are allowed at all, as the natural frequencies of the blade do not need to be modified separately. Masses are instead applied as load frame masses. Springs for uniaxial testing are also only allowed in the respective test direction to raise the corresponding natural frequency.

### 5     3.3 Case study rotor blade

In this work, a modern industrial rotor blade is used to demonstrate the developed method[1]. The target bending moment distribution is given and the area of interest is defined between 14% and 55% of the blade length. In this instance, the customer had previously defined the area of interest. Prior to testing, the blade tip was cut off at 90% of the blade length in order to reduce aerodynamic damping. The uniaxial tests are designed for 2.5 million load cycles in the flapwise direction

and 5.0 million cycles in lead-lag direction, respectively.

As the biaxial test is performed elliptically, the same number of load cycles is required for both test directions. Therefore, the required number of load cycles for the biaxial test is defined as 5.0 million and the target load level for the flapwise direction is reduced to 94.25% of the uniaxial load level. This is done according to the international standard IEC-61400-23.

The possible positions for the load introduction are given in Table 1. These positions are defined in the test specification and

thus are assumed appropriate for this purpose, and not to include critical regions of the blade. Other positions may contain critical areas, which need to be tested and must not be disturbed by load introduction equipment. Table 1 also contains the maximum spring element stiffness and the maximum mass at the corresponding position. These values are used to constrain the design space of the optimizations.

In the optimization for this blade, virtual masses or spring elements were only allowed up to the load position at 62% of the

blade length. At 76% and 88% of the blade length, only load frame masses directly attached to the blade have been permitted. At these positions, the deflection of the blade is too high to reasonably consider virtual masses or spring elements.

**Table 1: Load positions and boundary conditions for load element**

| Position [% of blade length] | 26 | 38 | 48 | 62 | 76 | 88 |
|---|---|---|---|---|---|---|
| Maximum stiffness [kN/m] | 100 | 70 | 55 | 45 | - | - |
| Maximum mass [t] | 10.0 | 7.0 | 5.5 | 4.5 | 1.5 | 1.0 |

### 25     3.4 Case study optimizations

The described method was used to generate three different test setups, including two state-of-the-art uniaxial setups for separate flapwise and lead-lag tests, which are used as a baseline in the comparison. The uniaxial setups were then optimized

---

[1] Detailed information on the blade cannot be provided due to confidentiality constraints



again, allowing springs to accelerate the tests. Additionally, one biaxial setup was generated using springs and virtual masses. Finding a suitable biaxial setup was much more challenging for the optimizer in terms of computational effort, as there were more constraints to consider and more design parameters to change. The final setups generated by the optimizer were subsequently evaluated with nonlinear transient analyses, to check whether nonlinearities resulted in different outcomes

5   than the fast harmonic simulation. These only showed minor differences for the given test setups.

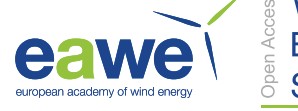
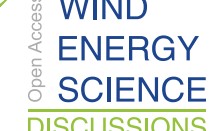

# 4 Results

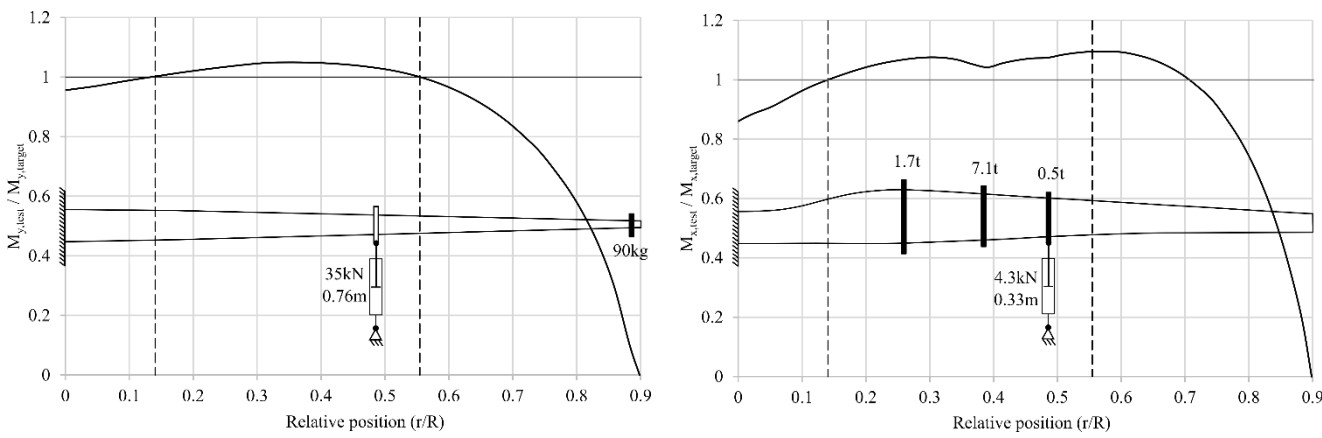

**Figure 4: Schematic of uniaxial test setups and test/target load ratio in flapwise (left) and lead-lag (right) directions**

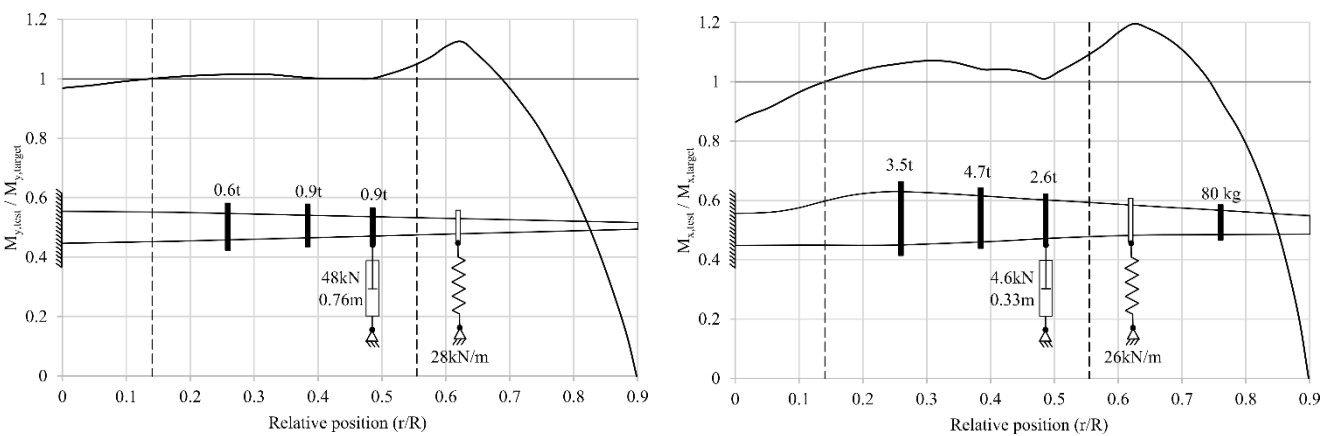

5 **Figure 5: Schematic of setups with springs and test/target load ratio in flapwise (left) and lead-lag (right) directions**

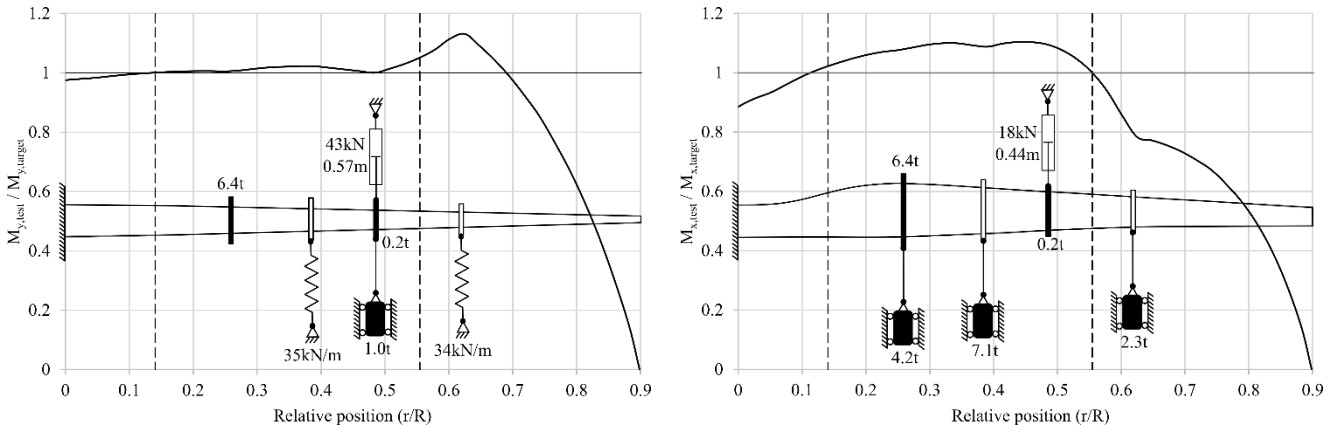

**Figure 6: Schematic of biaxial setup and test/target load ratio in flapwise (left) and lead-lag (right) directions**





Figures 4 to 6 show the schematics of the setups that were found by the different optimization runs. In addition, the corresponding ratio between actual moment distribution and target distribution (flapwise direction: $M_{y,test}/M_{y,target}$; lead-lag direction: $M_{x,test}/M_{x,target}$) over the normalized position along the rotor blade (r/R) is shown. The borders of the area of interest are marked with dashed lines. Table 2 summarizes the chosen design parameters and corresponding results of the found

5 setups. For all setups, the actuator was positioned at the third load frame at 48% of the blade length. This appears to be the position at which actuator force and stroke are balanced. Further outboard the stroke becomes too large, while further inboard the forces are too large, particularly in the flapwise direction. The optimizer only used a few springs but otherwise favoured masses for the uniaxial setups, even when springs were permitted at the first four load frame positions. Additional springs would have a negative effect on the moment distribution.

10 **Table 2: Summary of the optimized setups**

| Optimization results | Configuration | | Uniaxial | | Uniaxial (spring) | | Biaxial | |
|---|---|---|---|---|---|---|---|---|
| | Direction | | Flapwise | Lead-Lag | Flapwise | Lead-Lag | Flapwise | Lead-Lag |
| Chosen design parameters | Actuator Position | | 48% | 48% | 48% | 48% | 48% | |
| | Load frame mass [t] | at 26% | - | 1.7 | 0.6 | 3.5 | 6.4 | |
| | | at 38% | - | 7.1 | 0.9 | 4.7 | - | |
| | | at 48% | - | 0.5 | 0.9 | 2.6 | 0.2 | |
| | | at 62% | - | - | - | - | - | |
| | | at 76% | - | - | - | 0.08 | - | |
| | | at 88% | 0.09 | - | - | - | - | |
| | Virtual mass [t] | at 26% | N/A | N/A | N/A | N/A | - | 4.2 |
| | | at 38% | N/A | N/A | N/A | N/A | - | 7.1 |
| | | at 48% | N/A | N/A | N/A | N/A | 1.0 | - |
| | | at 62% | N/A | N/A | N/A | N/A | - | 2.3 |
| | Spring stiffness [kN/m] | at 26% | N/A | N/A | - | - | - | - |
| | | at 38% | N/A | N/A | - | - | 34.9 | - |
| | | at 48% | N/A | N/A | - | - | - | - |
| | | at 62% | N/A | N/A | 27.6 | 25.8 | 34.2 | - |
| Target and constraint results | Frequency [Hz] | | 0.574 | 0.789 | 0.666 | 0.807 | 0.680 | |
| | Test duration [days] | | 50.4 | 73.4 | 43.4 | 71.7 | 85.1 | |
| | Actuator stroke [m] | | 0.76 | 0.33 | 0.76 | 0.33 | 0.57 | 0.44 |
| | Actuator force amplitude [kN] | | 34.6 | 4.3 | 47.8 | 4.6 | 43.4 | 18.1 |
| | Maximum relative load within area of interest | | 1.049 | 1.095 | 1.050 | 1.092 | 1.051 | 1.104 |





### 4.1 Comparison

A comparison of the load distribution of the different test setups, as seen in Fig. 7, shows that the solutions with springs show higher overloads outside the area of interest, where it is permitted. This is due to stiff springs which are attached outboard of the test area at 62% of the blade length to raise the eigenfrequency as much as possible. For the flapwise setup, this is seen in both uniaxial and biaxial. For the lead-lag direction, it is only seen for the uniaxial test, as the lead-lag eigenfrequency for the biaxial test had to be lowered to match the flapwise frequency. It should be considered to reinforce the overloaded area around 62% of the blade length in order to prevent damages and expensive repairs during testing. It could also be possible to distribute the spring force into two separate springs that are close to each other, but at different cross-sections. Using less stiff springs could also be a possibility, but would lead to lower frequencies and longer test times.

The biaxial setup also exceeds the allowed lead-lag load by 0.4% around 45% of the blade length, as the optimizer was unable to find a better solution. As this is still very close to the allowable range, the overload is considered to be acceptable. Overall, the test quality in terms of load distribution within the area of interest is similar and reasonable for all test setups.

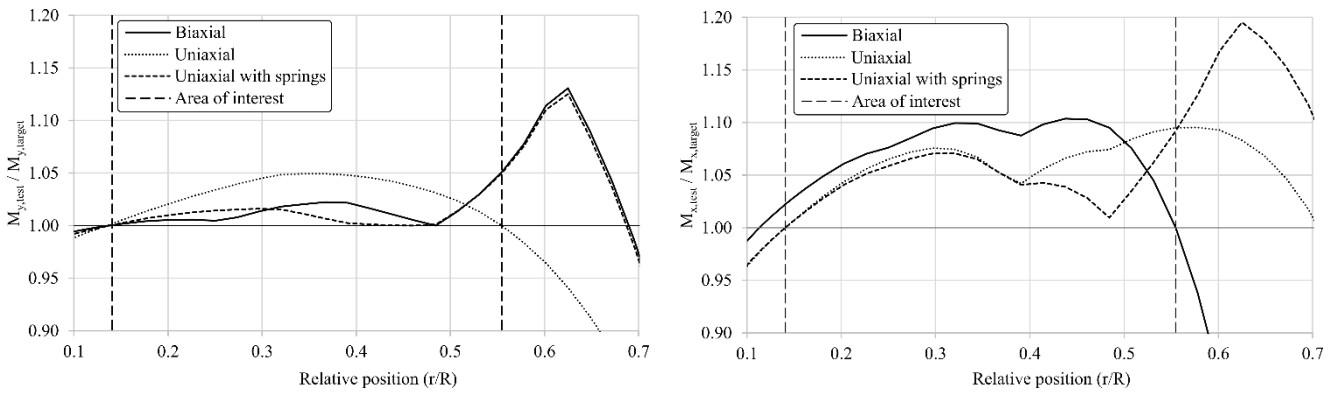

**Figure 7: Test load distribution relative to target load for different test setups in flapwise (left) and lead-lag (right) directions.**

A comparison of the time required for the tests using the different setups is shown in Fig. 8. The test times shown only take into account the active testing time; setup, inspections and trial runs are not considered. They are derived from the test frequency and the required number of cycles. The total time required by both consecutive uniaxial tests without springs is used as the baseline reference for the comparison, i.e. 100%. As the uniaxial lead-lag test needs twice as many cycles as the uniaxial flapwise test, it requires more time, even though the frequency is higher. When adding springs to the uniaxial tests, it is possible to reduce the total test time by 7%. Changing to biaxial tests can save an additional 24% of test time compared to the uniaxial test with springs.



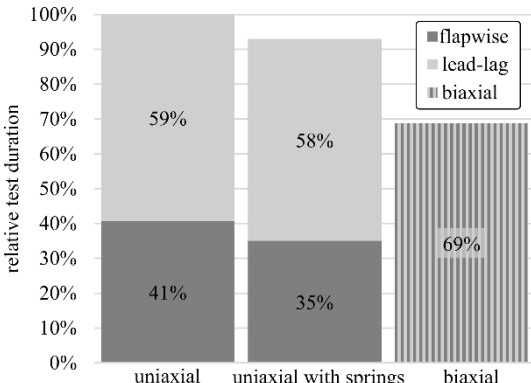

**Figure 8: Relative comparison of test durations.**



## 5 Summary and Outlook

In this paper, an optimization scheme to find test setups for fatigue testing of wind turbine blades was presented. Virtual masses and spring elements were implemented to tune the dynamic response of the test system. A case study was done on a representative blade. Results show that the described optimization method is suitable to find both uniaxial and biaxial test

setups.

For uniaxial test setups without spring elements, the optimization yielded results that satisfy all given boundary conditions in terms of required load distribution and local shear forces, yielding test frequencies of 0.57 Hz (flapwise) and 0.79 Hz (lead-lag direction). Compared to test setups that were defined manually, this is already an improvement in terms of testing time and overloading.

Furthermore, it was shown that an elliptical biaxial blade fatigue test using virtual masses and spring elements can save approximately 30% of testing time relative to consecutive state-of-the-art uniaxial fatigue tests.

As the results show, particularly for the flapwise test direction, the required excitation forces are near the specified limit of 50 kN. This occurs due to high aerodynamic damping resulting from the flapwise motion. Combining the biaxial test with aerodynamic fairings, as proposed by Pan et al. (2017), the damping may be reduced and less force would be needed.

Another approach to reduce the damping is to cut off more of the blade tip and combine biaxial with segmented testing. A further benefit of this would be the mass reduction, thus raising natural frequencies and accelerating the test even further. In addition, the first flapwise and lead-lag natural frequencies of the root segment are closer to each other than those of the whole blade. This way, less virtual elements may be needed to get the natural frequencies closer together to achieve elliptical biaxial testing.

Further future work includes extending the comparison to off-axis load distribution of the test to the target loads, not only the flapwise and lead-lag loads. This will show how much closer the applied loads in biaxial testing are to the field conditions as compared to uniaxial tests.



**Code/Data availability**

The data that supports the finding of this research are not publicly available due to confidentiality constraints.

**Author contribution**

DM, BM and SN conceptualized and defined requirements for the developed method. BM supervised the work. DM

5   developed the model code and performed the simulations. DM prepared the manuscript with contributions from all co-authors.

**Competing interests**

The authors declare that they have no conflict of interest.

**Acknowledgements**

10   We acknowledge the support within the Future Concept Fatigue Strength of Rotor Blades project granted by the German Federal Ministry for Economic Affairs and Energy (BMWi) (0325939) and the The Senator for Economic Affairs, Labour and Europe of the Free Hanseatic City of Bremen within the ERDF programme Bremen 2014-2020 (201/PF_IWES_ZK_Phase I/2017).





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
