# Peer review of "A novel rotor blade fatigue test setup with elliptical biaxial resonant excitation"

_Wind Energy Science, 2019_

## Referee Comment (RC1) · Peter Greaves (Referee) · 14 Jan 2020

I think this is a very interesting piece of work - I tried something similar to this during my PhD, but using much less robust methods. I found that the masses which were required seemed very high to me at the time, but I didn't appreciate that there are methods of applying these virtual masses which do not require them to be at the same height as the blade.

This test method has the advantage over the test optimisation method being pursued by ourselves and others (in which the frequencies do not need to coincide) that much less information sharing needs to take place between test house and customer, but perhaps at the expense of a more challenging test set-up process.

[Figure]

I think it is very important that you validated your results with a nonlinear time-stepping simulation as the nonlinearities could lead to significant angular changes of the push rods.

For the iteration to obtain the aerodynamic loads, I have considered a different way of doing this which may be of interest:

1 - Scale the mode shape so that the test loads match the target loads in a least squared sense 2 - Calculate energy dissipated during cycle by aerodynamic and structural damping (using a damping matrix generated by Rayleigh method) 3 - Use the actuator displacement to calculate the actuator force by equating the energy in (integral of actuator force x distance over cycle) to the energy out (air resistance and structural damping)

If it is possible, it would be very interesting to know a rough size range for the blade (I appreciate you can't share the exact length as this can identify the blade) as this would help contextualize the magnitude of the loads and masses required.

Overall, congratulations on a great piece of work!

---

## Referee Comment (RC2) · Nathan Post (Referee) · 3 Feb 2020

**General comments:**

This paper summarizes some key aspects of performing resonance fatigue tests of large wind turbine blades and the challenges faced when considering biaxial testing of these blades. The new work presented in the paper is applying a 3-dimensional harmonic model to evaluating and designing a fatigue test. However, the information disclosed regarding the implementation of the model is insufficient to enable another researcher to implement this approach directly. It is discussed that scaling the deflection mode shape is performed but not clear how this is accomplished in the biaxial case. Also, the tendency of a typical blade to not have perpendicular movement in the

flap and lead-lag directions due to the twist and relative frequencies is not addressed. While the actuators might be placed at angles, the blade motion might also be at an angle. Finally, no mention of incorporating the bend twist coupling in the model is addressed. Thus, the authors fail to address or take advantage of any potential benefits of the proposed 3D modelling approach – reserving that for future work. So, they have added complexity to the model while not demonstrating the superiority or even the difference between this approach when compared to performing the 2 simultaneous 2D harmonic models employed by Post et al. (Post 2016).

The use of spring elements is suggested – however, it is not obvious how such springs could be implemented effectively on a test since in this application they are subjected to reversing load cycles and most typical long displacement springs are either compression or tension, not both. Also withstanding the number of load cycles could be difficult. There is a brief discussion of the actuator displacements in a skew coordinate system which the reader assumes is used in the simulation (are they taken as displacement actuators rather than force actuators in the simulation?). It is not clear if the controls in the simulation assume contribution of each actuator in each direction. Since the change in angles of the actuators with displacement is neglected in the model it isn't clear what information is gained in this part of the analysis rather than just setting up the actuators to be perpendicular in the test.

Validation of the results was not conducted experimentally, nor were the results compared to previous simulation approaches in a rigorous way. A note that the resulting moments are within 3

**Specific comments:**

Page 3, 1 – 10. Reference is made to spring elements in the context of Post 2016. However, that report does not discuss the use of spring elements and instead uses the concept of negative virtual masses created with a hydraulic actuator to "remove" or carry mass from the blade and load frames. While the effect is similar in that both

a spring or a negative virtual mass provide a force in the opposing and proportional to the displacement (and the equivalent spring contestant $k = -mA(2\pi f)^2$ for a virtual mass with negative value of $m$ with displacement amplitude $A$ and frequency $f$) this is not a discussion that is included in Post 2016. In that report the authors discuss using actuators to remove the effect of mass thus introduction negative virtual masses into the test design. Recommend rewriting these paragraphs to accurately paraphrase the Post 2016 report and then introduce the concept of springs and the associated math separately.

Page 4, lines 19-24. This part of the paragraph doesn't make sense to me and I am not sure what the authors are trying to convey. How does the blade oscillate in different directions? Are we talking about for a uniaxial test or a biaxial test? The sentence "The effect of an element on the eigenfrequency, which is not to be affected, shall be minimized." makes no sense to me. Each element of the blade or saddle, mass, virtual mass or spring will change the eigenfrequency. Also, it isn't clear how this leads to the following sentence that the elements (which elements? load elements?) must be perpendicular to the mode shape of the blade. And what is not to be affected? I take it that you are trying to say that the load element vectors should be perpendicular to the local movement of the blade in the other mode-shape so as not to impart energy in that direction? I think that this relationship might influence the phase angle of the test and relative amplitude of the directions, but it is unclear how it would significantly impact the frequency or mode shape. Also, the actuators will be of finite lengths so the angles will change throughout the test and thus will impart some virtual mass effect in the perpendicular direction. Finally, the skew of the actuators discussed on page 5 seems to go counter to the argument made here.

Page 5, line 7 and 8. The authors state "The phase angle needs to be controlled during the test therefore, the hydraulic actuators need to be attached at the same position along the blade length". What is the reason for this? It is not clear to this reviewer that this statement is true. While you do need to control the phase angle, this is controlled

with the relative phase of the excitation of each actuator. The blade will move in its mode shape and phase angle regardless of where each exciter is placed along the blade length.

Page 6, line 2-3. Neglecting the non-linear displacement seems like a large oversight given the 3D model. Are the actuators force or displacement actuators? Depending on how significant the angles are and recognizing that for an elliptical test with a 90 deg phase angle, the maximum force of the actuator occurs at maximum angle it seems like this could be a significant loss in test efficiency and thus greater than simulated forces would be required in reality to run the test. Suggest expanding this discussion and better highlighting the impacts of the assumptions made and how the forces are introduced in the simulation.

Page 6, line 29-30: For a biaxial test, isn't the objective to modify the flap and lead-lag frequencies to be the same (1:1 test) so it isn't clear why they are different to start with. Do you mean that you are taking the mean of the uniaxial test cases as the guess for starting the biaxial test case?

Page 7, lines 4-8: While the iteration on the damping is included it isn't clear how this process adjusts the masses and springs to achieve the same frequency in both mode shapes for the biaxial test. At some point you are optimizing for maximum frequency within the bending moment limits but again it isn't clear how this is performed for the biaxial test while keeping the frequency in the flap and lead-lag directions the same. A flow chart or itemized list of steps of the simulation and optimization process would be helpful to clarify when each step is performed and what the objective functions are for each step.

Page 8, lines 10-11: As mentioned previously this comparison of the model results to the transient test (and how the transient test was constructed) would be good to include here (or later when comparing results in which case don't mention it hear but do describe the other simulations that you compare the results to. Also a comparison

to a simpler 2D harmonic approach would be very interesting as well and make this a stronger paper.

Page 8, line 17: spring elements are assumed to be massless? It is unclear how this would be accomplished. At a minimum a load frame is required to introduce the load to the blade from the spring and real springs do have mass so this seems like a gross oversimplification when designing the test.

Page 13 line 2: Allowing higher overloads outside of test regions is definitely something that would need to be taken on with care. Maybe if there is significantly more safety factor in that region of the blade it would be ok but it would be surprising if this is in generally reasonable. Same with reinforcing the blade in those regions – which will be difficult to do without creating stress concentrations.

Page 13, line 10. How did the optimizer end up exceeding one of the constraints? This needs to be explained since it should have found a solution within the constraints imposed, right? While this might be the "best" test solution for the blade, it isn't clear how the would have gone there without the user allowing it.

**Technical corrections:**

Page 2, line 14: "In order to safely proceed testing . . ." should be "In order to safely proceed with testing"

Page 3, line 3: Reference (Post, 2016) is not included in the list of references at the end of the paper.

Page 5, line 18: "Angle of attack" isn't a term that makes sense here since we generally think of that as an aerodynamic term. I think you mean the angle of incidence to the blade (or loadframe) – the alpha and beta in Figure 3. Suggest rewording this.

---

## Author Comment (AC1) · 13 Mar 2020

Referee comment:
I think this is a very interesting piece of work - I tried something similar to this during my PhD, but using much less robust methods. I found that the masses which were required seemed very high to me at the time, but I didn't appreciate that there are methods of applying these virtual masses which do not require them to be at the same height as the blade. This test method has the advantage over the test optimisation method being pursued by ourselves and others (in which the frequencies do not need to coincide) that much less information sharing needs to take place between test house and customer, but perhaps at the expense of a more challenging test set-up process.

Author response:
Thank you for the appreciative comments.
——————————————-

Referee comment:
I think it is very important that you validated your results with a nonlinear time-stepping simulation as the nonlinearities could lead to significant angular changes of the push rods.

Author response:
A nonlinear time-stepping simulation has been performed as mentioned on page 7 line 9-11 and page 10 line 3-5. The angular changes of the pushrods occurred, but did not significantly change the results compared to the harmonic simulation. The angular changes also depend significantly on the length of the pushrods. The longer the pushrods, the smaller the angular changes. At outboard positions with high deflections the pushrods would need to be unreasonably long to prevent high angular changes. Hence, at the most outboard positions no load elements requiring pushrods were allowed in the test desing.

Corresponding change #1:
Page 7 line 9: A transient simulation of the test, **considering nonlinear damping and nonlinear geometric effects, such as angular changes of load introduction elements**, requires about 100 times more processing time. A comparison of transient and harmonic simulations yielded less than 3% difference in loading amplitudes within the area of interest. **Hence, the harmonic results are sufficiently precise. Nevertheless, the final test setup needs to be checked by a transient simulation.**
——————————————-
Referee comment:
For the iteration to obtain the aerodynamic loads, I have considered a different way of

doing this which may be of interest: 1 - Scale the mode shape so that the test loads match the target loads in a least squared sense 2 - Calculate energy dissipated during cycle by aerodynamic and structural damping (using a damping matrix generated by Rayleigh method) 3 - Use the actuator displacement to calculate the actuator force by equating the energy in (integral of actuator force x distance over cycle) to the energy out (air resistance and structural damping)

Author response:
Interesting method, but as I understand this only helps to predict the required actuator force. The effect of aerodynamic damping on the load distribution cannot be predicted, which was the goal in the present work.
—————————————————-
Referee comment:
If it is possible, it would be very interesting to know a rough size range for the blade (I appreciate you can't share the exact length as this can identify the blade) as this would help contextualize the magnitude of the loads and masses required.

Author response:
The Blade length is above 60m.

Corresponding change #2:
Page 7 line 9: In this work, a modern industrial rotor blade **of more than 60m in length** is used to demonstrate the developed method.
—————————————————-
Referee comment:
Overall, congratulations on a great piece of work!

Author response:
Thank you for your helpful comments!

---

## Author Comment (AC2) · 13 Mar 2020

First of all thank you a lot for the numerous helpful comments!

**General comment:**

Referee comment:
This paper summarizes some key aspects of performing resonance fatigue tests of large wind turbine blades and the challenges faced when considering biaxial testing of these blades. The new work presented in the paper is applying a 3-dimensional harmonic model to evaluating and designing a fatigue test. However, the information

disclosed regarding the implementation of the model is insufficient to enable another researcher to implement this approach directly. It is discussed that scaling the deflection mode shape is performed but not clear how this is accomplished in the biaxial case.

Author response:
The mode shapes are not scaled, but the displacement excitation of the harmonic simulation. The harmonic simulation for biaxial testing is also not necessarily exciting at a natural frequency of the system.
The deflection is realised by displacement excitation using two actuators. The biaxial simulation is scaled iteratively. First a small displacement is applied in both actuators at the same time using a phase difference as described on page 5. After evaluating the load distribution two scaling factors for the flapwise and lead-lag load are computed. These are multiplied with the corresponding blade deflection at excitation position. The actuator stroke for the next simulation is then computed using eq. (1) and eq. (2). This scaling procedure is repeated until convergence.

Corresponding change #3:
Page 7 line 1: The test is then simulated using a harmonic simulation with a **small** initial actuator **displacement** as excitation.
Page 7 line 3: …to meet test requirements. **In the biaxial case, the excitations of both actuators are applied simultaneously. Hereby, separate scaling factors are applied to flapwise and lead-lag displacement, which are converted to actuator excitations using Eq. (1) and (2) before repeating the harmonic simulation.**
Page 7 line 5: This iterative process of updating the damping and the **scaled** harmonic analysis is repeated until convergence is achieved.

Referee comment:
Also, the tendency of a typical blade to not have perpendicular movement in the flap and lead-lag directions due to the twist and relative frequencies is not addressed.

While the actuators might be placed at angles, the blade motion might also be at an angle.

Author response:
This tendency of the blade is rather key element of the load element procedure. This is mentioned on page 4 line 18: "as a rotor blade oscillates in one of its first mode shapes, the direction of displacement is not necessarily aligned with the main directions of the blade..." The not-perpendicular movement is also displayed in Figure 2(a).

Corresponding change #4:
Page 4 line 19: Additionally, the mode shape direction changes along the length of the blade, meaning every cross section may oscillate in a different direction. **The first flapwise and lead-lag mode shape directions are not perpendicular to each other.** The effect of a flapwise element on the **lead-lag natural frequency and vice versa** shall be minimized. Hence, it was found that the elements need to be attached perpendicular to the mode shape of the rotor blades, whose frequency is not to be affected.

Referee comment:
Finally, no mention of incorporating the bend twist coupling in the model is addressed.

Author response: The bend twist coupling is not mentioned specifically, but the general coupling of different degrees of freedom (if applicable) is taken into account in the simulations as described in section 2.1.

Corresponding change #5:
Page 4 line 9: These properties, **which consist of mass matrices and fully populated 6x6 stiffness matrices**, are derived from preceding analyses of multiple

different cross sections along the blade.

Referee comment:
So, they have added complexity to the model while not demonstrating the superiority or even the difference between this approach when compared to performing the 2 simultaneous 2D harmonic models employed by Post et al. (Post 2016).

Author response:
By performing two simultaneous 2D simulations as employed by Post et al. (2016) the flapwise and lead-lag loading can be evaluated independently very well. But when considering coupled mode shapes, load introduction elements with tilted angles of inclination, and multiple non-perpendicular excitations at different phase angles only 3D simulations are applicable. Since the goal of this work was to incorporate all of these effects the superposition of 2 simultaneous 2D simulations would not suffice.

Corresponding change #6:
Page 3 line 11: Instead of two separate two-dimensional simulations one three-dimensional simulation is performed. **This way the coupling of multiple load and deflection directions can be considered.** Here, the virtual masses and spring elements are not just applied in the global flapwise or lead-lag direction, but tilted around the blade axis as described below.
————————————————-
Referee comment:
The use of spring elements is suggested – however, it is not obvious how such springs could be implemented effectively on a test since in this application they are subjected to reversing load cycles and most typical long displacement springs are either compression or tension, not both. Also withstanding the number of load cycles could be difficult.

Author response:

Since the current work is a numeric study, tension-compression-spring elements were used in the simulation. How these springs are to be realized in reality is not part of this work. However, this aspect is under investigation and such information will be available in the near future.
————————————-

Referee comment:
There is a brief discussion of the actuator displacements in a skew coordinate system which the reader assumes is used in the simulation (are they taken as displacement actuators rather than force actuators in the simulation?). It is not clear if the controls in the simulation assume contribution of each actuator in each direction. Since the change in angles of the actuators with displacement is neglected in the model it isn't clear what information is gained in this part of the analysis rather than just setting up the actuators to be perpendicular in the test.

Author response:
Yes, the excitation is implemented displacement driven. Since the actuators are not perpendicular and at angles both actuators contribute to both directions.

Corresponding change #7:
Page 5 line 8: As the hydraulic actuators may not be in line with the main directions of the blade **and not perpendicular to each other**, a skew coordinate system can be derived from their orientation as shown in Fig. 3.
Page 5 line 13: In order to find the correct **displacement** excitation for the actuators, the desired blade motion, including the phase difference, must be converted to the skew coordinate system using Eq. (1) and Eq. (2).
————————————-

Referee comment:
Validation of the results was not conducted experimentally, nor were the results compared to previous simulation approaches in a rigorous way.

Author response:
The uniaxial simulation procedure was validated using confidential data of different blades. Experimental results for validation of biaxial simulations are not available at this stage. The comparison to previous biaxial simulation approaches is hardly possible since no published approaches consider 3D-motion and fully populated stiffness matrices.
Also further comparisons to experiments and other simulations would go beyond the scope of this work.
————————————————

Referee comment:
A note that the resulting moments are within 3

Author response:
I believe there is a part of the comment missing
————————————————

**Specific comments:**

Referee comment:
Page 3, 1 – 10. Reference is made to spring elements in the context of Post 2016. However, that report does not discuss the use of spring elements and instead uses the concept of negative virtual masses created with a hydraulic actuator to "remove" or carry mass from the blade and load frames. While the effect is similar in that both a spring or a negative virtual mass provide a force in the opposing and proportional to the displacement (and the equivalent spring constant k=-mA$(2\pi f)^2$ a virtual mass with negative value of m with displacement amplitude A and frequency f) this is not a discussion that is included in Post 2016. In that report the authors discuss using actuators to remove the effect of mass thus introduction negative virtual masses into

the test design. Recommend rewriting these paragraphs to accurately paraphrase the Post 2016 report and then introduce the concept of springs and the associated math separately.

Author response:
Thank you for clarifying. This is changed accordingly:

Corresponding change #8:
Page 3 line 3: . . . with the concept of virtual masses (Post, 2016). These elements are connected to the blade in such a manner, that they only add **or subtract** inertia in one **specific** direction.
Page 3 line 7: To adjust the natural frequencies to be equal or very close to each other, i.e. raising the flapwise and/or lowering the lead-lag natural frequency, virtual masses are used as Post et al. (2016) suggested. **Rather than using negative virtual masses to remove inertia from the system, here spring elements are used to add stiffness. Adding stiffness or removing inertia at a specific position have physically a similar effect on the systems natural frequencies.**
————————————-
Referee comment:
Page 4, lines 19-24. This part of the paragraph doesn't make sense to me and I am not sure what the authors are trying to convey. How does the blade oscillate in different directions?
Are we talking about for a uniaxial test or a biaxial test?
The sentence "The effect of an element on the eigenfrequency, which is not to be affected, shall be minimized." makes no sense to me. Each element of the blade or saddle, mass, virtual mass or spring will change the eigenfrequency. Also, it isn't clear how this leads to the following sentence that the elements (which elements? load elements?) must be perpendicular to the mode shape of the blade. And what is not to be affected? I take it that you are trying to say that the load element vectors should be

perpendicular to the local movement of the blade in the other mode-shape so as not to impart energy in that direction? I think that this relationship might influence the phase angle of the test and relative amplitude of the directions, but it is unclear how it would significantly impact the frequency or mode shape.

Author response:
Not the blade oscillates in different direction but the oscillation direction changes slightly along the blade. Hence, the motion near the root has a slightly different direction than the motion at the tip.
We are talking about general mode shapes of the blade, not about testing.
Since the first flapwise and first lead-lag mode shapes are not perpendicular, the load element direction needs to be defined more specifically. In order to affect mainly the first flapwise frequency by a flapwise load element and only in a negligible amount the first lead-lag frequency, the element is oriented perpendicular to the corresponding lead-lag mode shape direction, rather than in line with the flapwise mode shape direction. Vice versa for a lead-lag load element.

Corresponding change #9:
Page 4 line 19: Additionally, the **direction of a single** mode shape changes along the length of the blade **by a few degrees**, meaning every cross section may oscillate in a **slightly** different direction.
Also, see Changes #3

Referee comment:
Also, the actuators will be of finite lengths so the angles will change throughout the test and thus will impart some virtual mass effect in the perpendicular direction.

Author response:
This is true, but using the described procedure of orienting the load elements this

effect is reduced to a minimum

Referee comment:
Finally, the skew of the actuators discussed on page 5 seems to go counter to the argument made here.

Author response:
The actuators are supposed to excite the test and not affect the natural frequencies. Due to the applied springs and virtual masses the mode shapes of the total test setup differ from the mode shapes of the blade. In order to excite the test in these new mode shapes the actuators are applied accordingly.
————————————

Referee comment:
Page 5, line 7 and 8. The authors state "The phase angle needs to be controlled during the test therefore, the hydraulic actuators need to be attached at the same position along the blade length". What is the reason for this? It is not clear to this reviewer that this statement is true. While you do need to control the phase angle, this is controlled with the relative phase of the excitation of each actuator. The blade will move in its mode shape and phase angle regardless of where each exciter is placed along the blade length.

Author response:
The phase between the flapwise and lead-lag motion depends purely on the phase between the actuator displacements. When placing the actuators at the same position the phase of flapwise and lead-lag motion at this position directly corresponds to the actuator phase. When placing the actuators at different positions instead, it is harder to control the phase angle of the blade. This would require a complex controller, since the phase of motion at different cross section varies and depends on the mass and stiffness distribution. Using the same phase of excitation at different positions would

result in different blade movements.

Corresponding change #10:
Page 5 line 7: This phase **shift** needs to be **adjusted** during testing. **If both** hydraulic actuators are attached at the same position along the blade length, **the phase shift between the flapwise and lead-lag motion directly depends on the actuator phase shift. Otherwise a complex controller system would be needed, since the phase of motion varies along the blade length and the phase shift would then also depend on the blade properties. Therefore, the actuators are kept at the same position in this work.**

————————————————

Referee comment:
Page 6, line 2-3. Neglecting the non-linear displacement seems like a large oversight given the 3D model. Are the actuators force or displacement actuators? Depending on how significant the angles are and recognizing that for an elliptical test with a 90 deg phase angle, the maximum force of the actuator occurs at maximum angle it seems like this could be a significant loss in test efficiency and thus greater than simulated forces would be required in reality to run the test. Suggest expanding this discussion and better highlighting the impacts of the assumptions made and how the forces are introduced in the simulation.

Author response:
They are displacement actuators. Since harmonic simulations do not permit non-linerarities, neglecting them is inevitable. Though, this fast simulation method is only used to compare different test setups with each other in the design process. For this purpose mainly qualitative comparison is necessary. Hence, the harmonic simulation is still valid. When evaluating the final setup non-linear transient simulations are utilized to confirm the results.
Corresponding change #11:
Page 6 line 2: As harmonic simulations cannot consider nonlinear **effects** this phenomenon has been omitted **at this stage of the setup definition.**

————————————-

Referee comment:
Page 6, line 29-30: For a biaxial test, isn't the objective to modify the flap and lead-lag frequencies to be the same (1:1 test) so it isn't clear why they are different to start with. Do you mean that you are taking the mean of the uniaxial test cases as the guess for starting the biaxial test case?

Author response:
The two different frequencies are natural frequencies of the system, which are derived from the preceding modal analysis. The objective is to modify these natural frequencies to be the same or at lest close to each other. But in order to find such a test setup different setups need to be evaluated. Hence, the simulation uses a single forced excitation frequency which is between these natural frequencies.

Corresponding change #12:
Page 6 line 29: For a biaxial test, the mean value between first flapwise and first lead-lag **natural** frequency is used as the **forced** excitation frequency. **This way any given test setup can be evaluated without regard to the ratio between natural frequencies.**

————————————-

Referee comment:
Page 7, lines 4-8: While the iteration on the damping is included it isn't clear how this process adjusts the masses and springs to achieve the same frequency in both mode shapes for the biaxial test. At some point you are optimizing for maximum frequency within the bending moment limits but again it isn't clear how this is performed for the biaxial test while keeping the frequency in the flap and lead-lag directions the same. A

flow chart or itemized list of steps of the simulation and optimization process would be helpful to clarify when each step is performed and what the objective functions are for each step.

Author response:
The simulation procedure itself does not adjust the masses and springs. To achieve the same frequency the optimization described on page 8 is used to adjust these values.
The suggested flowchart is added to the work.

Corresponding change #13:
Additional Figure (see Fig 1.) with caption: **figure 4: Flow chart of simulation sequence and optimization**
Page 6 line 25: …given setup. **This sequence is summarised in Figure 4 on the left side.**
Page 8 line 4: …as described **in the following. This workflow is visualised in Figure 4.**
Page 8 line 13: Furthermore, the position of the actuators can be changed between the defined positions. **For the optimization of the biaxial test another constraint is introduced: In order for the excitation to be in resonance the first flapwise and lead-lag natural frequencies need to be the same or close to each other. Hence, a limit of 5% deviation between the first flapwise and lead-lag natural frequencies is accepted.**
————————————————-
Referee comment:
Page 7, lines 10-11: As mentioned previously this comparison of the model results to the transient test (and how the transient test was constructed) would be good to include here (or later when comparing results in which case don't mention it hear but do describe the other simulations that you compare the results to. Also a comparison

to a simpler 2D harmonic approach would be very interesting as well and make this a stronger paper.

Author response:
The detailed description of the transient simulation as well as comparison to other simulation approaches would go beyond the scope of this work. They are reserved for future work.
————————————————-
Referee comment:
Page 8, line 17: spring elements are assumed to be massless? It is unclear how this would be accomplished. At a minimum a load frame is required to introduce the load to the blade from the spring and real springs do have mass so this seems like a gross oversimplification when designing the test.

Author response:
When designing the spring elements in reality their stiffness would be chosen in such a way, that its own mass is compensated and the "active" stiffness delivered to the blade would be equal to the stiffness in the present case. This way the assumption of "massless" springs is valid.
————————————————-
Referee comment:
Page 13 line 2: Allowing higher overloads outside of test regions is definitely something that would need to be taken on with care. Maybe if there is significantly more safety factor in that region of the blade it would be ok but it would be surprising if this is in generally reasonable. Same with reinforcing the blade in those regions – which will be difficult to do without creating stress concentrations.

Author response:
the arguments in this comment are adopted

[Figure]

Corresponding change #14:
Page 13 line 6: ...lowered to match the flapwise frequency. **The overloaded area around 62% of the blade length would have to be further examined by the blade designer before testing in order to check if the design safety factors are high enough to withstand these loads. Otherwise** it should be considered to reinforce this part of the blade in order to prevent damages and expensive repairs during testing.
—————————————-

Referee comment:
Page 13, line 10. How did the optimizer end up exceeding one of the constraints? This needs to be explained since it should have found a solution within the constraints imposed, right? While this might be the "best" test solution for the blade, it isn't clear how the would have gone there without the user allowing it.

Author response:
The optimizer searches for the minimal value of the objective function, which consists of a sum of the actual objective and penalty values for constraint violations. If the algorithm is not able to find a design point which satisfies the constraints still the design point which gets the closest, while minimizing the actual objective, can be used as output of the optimization.
—————————————-

**Technical corrections:**

Referee comment:
Page 2, line 14: "In order to safely proceed testing : : :" should be "In order to safely proceed with testing"
Page 3, line 3: Reference (Post, 2016) is not included in the list of references at the end of the paper.

Page 5, line 18: "Angle of attack" isn't a term that makes sense here since we generally think of that as an aerodynamic term. I think you mean the angle of incidence to the blade (or loadframe) – the alpha and beta in Figure 3. Suggest rewording this.

Author response:
The technical corrections are applied as suggested

Corresponding change #15:
P2L14: In order to safely proceed **with** testing, these damages need. . .
P17L18: **Post, N., Bürckner, F.: Fatigue Test Design: Scenarios for Biaxial Fatigue Testing of a 60-Meter Wind Turbine Blade, Tech. rep., National Renewable Energy Laboratory, Golden, CO, USA, https://doi.org/10.2172/1271941, 2016.**
P5L18: In the real test, the angle of **incidence** of the actuators will change constantly as the blade follows the elliptical motion.
* * *
[Figure]

**Fig. 1.**